# Clean vs dirty labels: Transparency and authenticity of the labels of Ceylon cinnamon

Devarahandhi Achini Melda De Silva[1]*, Renda Kankanamge Chaturika Jeewanthi[1], Rajapakshage Heshani Navoda Rajapaksha[1], Weddagala Mudiyanselage Tharaka Bilindu Weddagala[1], Naoki Hirotsu[2], Bun-ichi Shimizu[2], Munasinghe Arachchige Jagath Priyantha Munasinghe[1]

1 Department of Agribusiness Management, Faculty of Agricultural Sciences, Sabaragamuwa University of Sri Lanka, Belihuloya, Sabaragamuwa Province, Sri Lanka, 2 Department of Life Sciences, Toyo University, Itakura, Gunma, Japan

☯ These authors contributed equally to this work.
* achini@agri.sab.ac.lk

**Data Availability Statement:** All relevant data are within the paper and its Supporting Information files.

## Abstract

Ceylon cinnamon, which was regarded as a luxury spice during ancient times, has been consumed for its medicinal properties and health benefits for thousands of years. For centuries, Arabian traders controlled the European cinnamon trade through limited supplies from a country which they did not reveal. Content marketing analysis and chemical profiling of value-added products of Ceylon cinnamon in the global marketplace are proposed to investigate the clean status of the product labels. In the present study, a mixed-method approach was employed to investigate the labels of 6 types of value-added forms of cinnamon; i.e. quills, powder, tea, breakfast cereals, confectionery and bakery and nutraceuticals which are used in USA, UK, Mexico, Japan and products of Sri Lankan cinnamon exporters. Two hundred and seventy-six labels were analyzed to find out the aspects of clean status, transparency and authenticity. Key label claims of the cinnamon products lie within the bounds of cleaner, healthy, nutritional and sustainable attributes. Consumer perception lies within ingredients, nutritional value, country of origin and claim on safety and quality standards and certification. The value chain transparency, ethical rules (species mislabeling), and chemical profile of the pharmaceutical, confectionery and fragrance industry inputs were ignored. The best claim and competitive advantage of the Ceylon cinnamon; an ultra-low level (<0.01 mg/g Dry Weight) of Coumarin, were rarely indicated in labels. Lack of clean labels and traceability lagged Ceylon cinnamon in the 40 international markets while Cassia cinnamon (Coumarin content 2.23 mg/g DW), a major competitor of Ceylon cinnamon appears in the market with dirty labels. Millennials and upper-middle-class female consumers in their active ages, place a high demand on Ceylon cinnamon. Today's tech-savvy global consumers of Ceylon cinnamon use market intelligence frequently for identifying product authenticity. Well equipped clean labels were found to be demanded by the modern cinnamon consumers.

**Funding:** The empirical research (market survey) informing this paper was funded by the Ministry of Social Welfare and Primary Industries of Sri Lanka and the National Science Foundation. DAM received the grant for project titled, Ceylon cinnamon value chain development: Making global market space for value chain actors (SP/CIN/2016/05). DAM and RKC received the research, innovation and commercialization grant of AHEAD (Accelerating Higher Education and Development) project for the development of novel beverages from Ceylon cinnamon (6026-LK/8743-LK/P159995) which facilitate the chemical profiling process. The funders had no role in study design, data collection and analysis, decision to publish or preparation of the manuscript.

**Competing interests:** The authors have declared that no competing interests exist.

# Introduction

Cinnamon has been one of the oldest, most popular and expensive baking spices in many cultures for centuries. Cinnamon trade of Sri Lanka dates back to the 14th century AD and a formal trade agreement between the Dutch Government and the King of Ceylon in 1766. Growing demand for healthy food products has led to an upsurge in the demand for cinnamon in the North American and European markets. Attributed mainly to its various anti-fungal, anti-viral, anti-carcinogenic and anti-bacterial properties, cinnamon is commonly used in the manufacture of food (confectionery and bakery), beverage, medicine (pharmaceutical and nutraceutical) and cosmetics which are witnessing significant positive demand. Moreover, global demand for natural food and beverages without jeopardizing shelf life, nutritional quality and taste are increasing positively [1]. Global food and beverage manufacturers have identified cinnamon; traditional additive and preservative, as the best replacement against artificial ingredients. Cinnamon has been recognized as a commercially available natural sweetener, flavouring agent, additive and preservative. Antioxidant, antifungal and antibacterial properties of cinnamon inhibit the deterioration of food and act as a natural food preservative. The mechanism of decreasing the DNA binding activity of the quorum sensing response regulator of LuxR grounds to the above properties claimed by the cinnamon [2]. Eugenol, which is the major component of cinnamon leaf oil, is a registered food additive that enhances flavour, especially on herbal products [3]. Linalool and β-Caryophyllene are registered as direct food additives even in synthetic forms [4], yet cinnamon provides them in a natural safe route to the consumer. Camphor from bark and roots of cinnamon is considered as a flavour enhancing compound in the food industry. The polyphenol content of cinnamon inhibits the lipid rancidity and reduces the deterioration rate. Incorporation of 3% cinnamon extract to butter resulted in low levels of peroxide value, free fatty acids value and low microbial count compared to the butter without a preservative and potassium sorbate added butter [5]. Antioxidant activity of cinnamon extends the shelf life of butter. Vidanagamage et al., suggested that cinnamon extract can be used to formulate an antioxidant-rich butter as a natural preservative for butter [5].

The essential oil of cinnamon has the potential to apply as a natural preservative in perishable food items such as fruits, fish and meat. Essential oil applications of true cinnamon have demonstrated its potential in inhibiting the mycelial growth of fungus in fruits and claimed as a preservative for perishable fruits [6]. Moreover, the food preservative properties of cinnamon have been investigated in fish at room temperature, fried fish, and deep-fried fish; as the activity index of the cinnamon is more than 0.5 [2]. In the same study, it has been proposed to coat the cinnamon paste in the inner package as a preservative. Ojagh et al., explained that the antioxidant mechanism of chitosan-based coating with cinnamon on fish enabled the good quality characteristics to be retained longer, extending the shelf life of fish [7] which encapsulated cinnamon essential oil into chitosan nanoparticles of three sizes [8]. These nanoparticles improved the antioxidant and antimicrobial properties of pork and acted as a natural preservative for meat. The nanoparticles with bigger size prolonged the shelf-life of the chilled pork by maintaining the quality of meat and meat products. The chitosan-based cinnamon coating on the defense-related enzymes, free radical scavenging activity, and the permeability and integrity of fruit cell membrane in fruits [9] are evidenced the suitability of applications of cinnamon as a natural preservative [10]. developed an anti-insect food packaging film containing cinnamon oil encapsulates that ensures the insect repelling properties of cinnamon without affecting the sensory qualities of the food.

The essential oil of true cinnamon has been examined for its preservative properties in bakery items as well. The effects of the cinnamon oil in cream-filled cakes as a natural preservative

in bakery industry especially for cream-filled cakes and pastries, have been identified [11]. Cinnamon improved the sensory quality of the cinnamon-based products while acting as a preservative. Kordsardouei et al., discussed the preservation effects of essential oil of true cinnamon for applications in cakes and suggested cinnamon as a powerful natural food preservative that can replace the synthetic preservatives and additives in foods [12]. Ceylon cinnamon has a higher potential to replace synthetic preservatives that are carcinogenic or toxic.

Due to the above advantages of true cinnamon, it is ethical and essential to differentiate the same from cassia in the industrial applications [13, 14]. Moreover, economic, social and environmental aspects of the cinnamon industry achieved sustainability for a long time. This is one of the major plantation industries which shares the harvest with its workforce. Harvest sharing varies from 33–50% between the grower/planter and the peeler [15].

East Asian cinnamon growing nations, such as Indonesia, China, and Vietnam control the lion's share of the global cinnamon market. Four Asian cinnamon producers supply different cinnamon varieties to the global market; Indonesian cinnamon (*Cinnamomum burmannii*) by Indonesia, Cassia cinnamon (*C. cassia*) by China, Saigon cinnamon (*C. loureiroi*) by Vietnam, and True cinnamon or Ceylon cinnamon (*C. zeylanicum*) by Sri Lanka. Indonesia leads the cinnamon exports by volume followed by China and the USA leads the global cinnamon imports followed by Germany (Fig 1A). Key items of Ceylon cinnamon export basket composed of bails and cut quills, and the global market share of which is less than 10%. Mexico followed by the USA is the leading Ceylon cinnamon importer (Fig 1B).

Ceylon cinnamon tree is a moderately sized, bushy, evergreen tree up to 16 m tall and 60 cm diameter. Buttresses up to 60 cm high, out 70 cm, thin. Cassia is a conical shaped small tree which maintains 3–4 m height under cultivation. Ceylon cinnamon branchlets are slender, compressed, glabrous, end bud partly finely silky, bright or light red. Leaves opposite or sub opposite, glabrous, oval or elliptic to lanceolate-oval or narrowly elliptic or broadly acuminate. Upper surface of the leaves is dark green, shining, smooth while the lower surface is paler. Leaves have 3 main prominent veins on both surfaces. Cassia leaves are opposite, glabrous above, minutely hairy below, hairs microscopic, three-ribbed from about 5 mm above the base, side veins ascending to the apex. Axillary panicle of Ceylon cinnamon is 20 cm long main peduncle and a few, stiff short branches while inflorescence is axillary panicle of Cassia, exceeding the leaves, many flowered, long peduncle, flowers with long pedicel and minutely hairy [16]. Pale yellowish green flowers and the dark purple fruit of both species have almost similar features (Fig 2). Colour and art of processing cinnamon quills help to distinguish the Ceylon cinnamon from Cassia. Cassia bark peels are dark brown-red coloured, double-sided rolls with a hollow structure, and thicker sticks of lumpier and tougher texture. Compared to the Ceylon cinnamon, Cassia has very strong pungency in flavour. Tan-brown coloured and smooth textured Ceylon cinnamon quill composed of tight thinner layers. After powdering the quills, it is very difficult to identify the differences between these two species without microscopic identification. Presence of a number of rounded sclereids as single or small groups in the microscopic observation is the prominent characteristic to identify Ceylon cinnamon following plentiful fibers with small uneven lumen and a lot of starch grains. Very rarely a traceable amount of needle-shaped crystals of calcium oxalate can be identified; however, cork fragments are absent or very rare. Microscopic identification of Cassia powder consists of a large amount of starch granules and greater fiber diameter with an average diameter 69.60 ± 8.85 μm. The average diameter 52.00 ± 5.25 μm of Ceylon cinnamon fiber provides an obvious physiochemical characteristic to differentiate it from Cassia powder [17].

Sri Lanka is renowned for centuries for producing and supplying true cinnamon to the world and, it currently holds 80% of the true cinnamon market. Ceylon cinnamon comes from the dried bark of a perennial tree, *C. zeylanicum*, native to Sri Lanka. Historical evidence

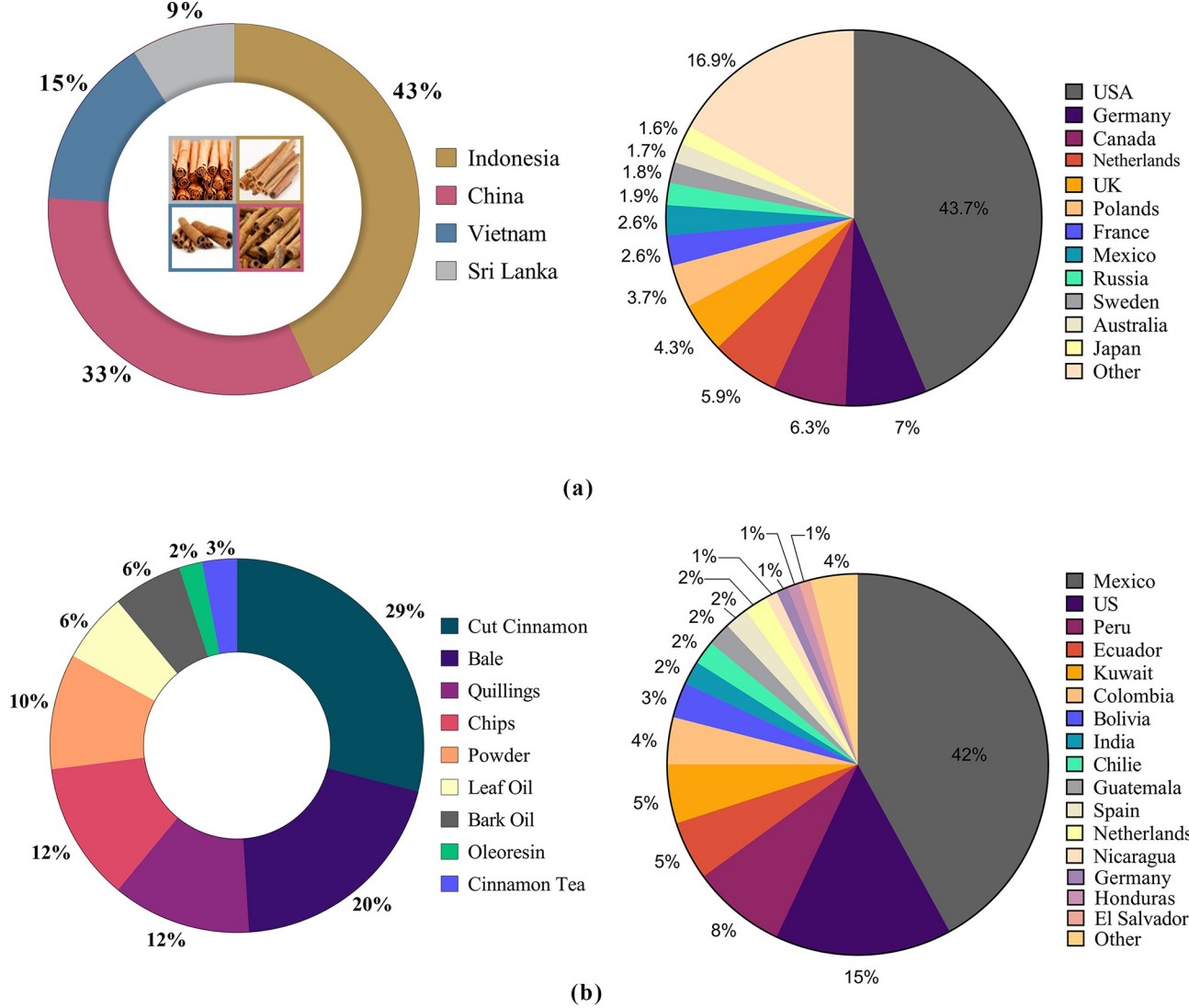

**Fig 1. Cinnamon market share & types.** (A): Market share (%) of the leading global cinnamon exporters & importers. Indonesia, holds the highest export market share, leads the global cinnamon exports followed by China. Sri Lanka, 4th largest supplier to the world cinnamon market while act as the leading supplier of true cinnamon (82%). USA dominated the cinnamon imports by placing the EU, historical leader into second place. (B): Composition of Ceylon cinnamon export basket & export destinations. Ceylon or true cinnamon exports cater both residential and commercial market segments globally. Mexico has emerged as one of the leading consumers of Ceylon cinnamon together with Latin America placing USA and EU into second place.

proved that Ceylon cinnamon was used as a natural preservative for meat in the 14th and 15th centuries. Quills were the first cinnamon products to be sold. The art of quill making is unique to Sri Lanka and the quill (quills are made by rolling of pieces of peeled barked joining together to get a pipe-like structure in 42' or 21' length) is currently not registered as a protected product. In contrast, the processing technique of Cassia cinnamon is different from true cinnamon and this helps consumers to separate Ceylon cinnamon from Cassia. Uses of cinnamon vary from region to region; USA and Canada use it as an ingredient of cooking, Latin America use in spice mixes, UK and EU use it for baking and Japan use it in chocolate and confectionery products.

However, Ceylon cinnamon's characteristic chemical properties have attracted a significant number of consumers of modern knowledge base society. More than 80 compounds were

*Cinnamomum zeylanicum*          *Cinnamomum cassia*

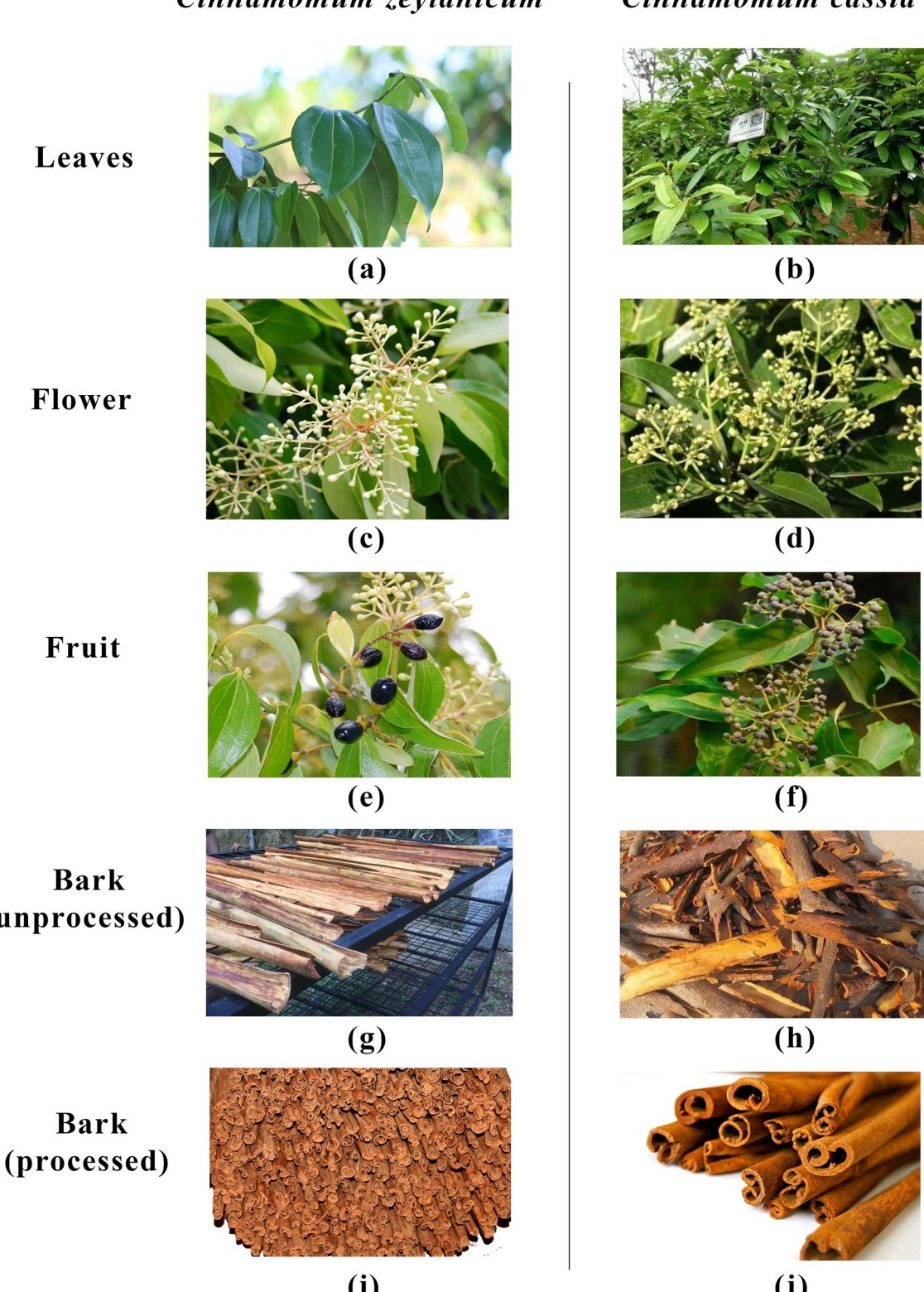

**Fig 2. The images of Cassia vs Ceylon cinnamon to compare the morphological characteristics.** Morphological characteristics of leaves, flowers, fruit, unprocessed bark and processed bark of *C. zeylanicum* Vs *C. cassia* is graphically compared.

identified from different parts of the cinnamon tree. The leaf oil has a major compound called eugenol which is present in higher levels in Ceylon cinnamon than in other *Cinnamomum* species [18, 19]

Ribeiro et al., analyzed the major compounds of leaf in *C. zeylanicum* and reported the presence of major compounds as Benzyl benzoate 74.2%, alpha-phellandrene 6.9%, alpha-pinene 3% and linalool 2.7% [19].

Cinnamaldehyde and camphor have been reported to be the major components of volatile oils from stem bark and root bark, respectively [20]. The chemical component of bark of *C. cassia* is reported as E-Cinnamaldehyde 92–98%, Z-cinnamaldehyde 0.8–2.7%, β-caryophyllene 0.4–3.6%, Coumarin 0.1–1.6% and α-ylangene 0.1–2.7%. Cinnamaldehyde level in Ceylon cinnamon is approximately 50–63% is the basis of its mild flavour compared to Cassia [21]. Proanthocyanidins-rich true cinnamon extract from the bark resulted in higher functional activities before the spray drying, freeze-drying, and concentrated processing. Total content of proanthocyanidins (mg eq procyanidin B2/g of extract dry basis) analysis showed the following; Folin-Ciocalteu reduction power (mg eq. Galic acid/g of extract dry basis), antioxidant capacity by DPPH and FRAP (mmol eq. Trolox/g of extract dry basis) and inhibitory activities ($IC_{50}$) of α-amylase and α-glucosidase (mg/ mL of reaction medium) for the Ceylon cinnamon extracts were 419 ± 4, 740 ± 8, 2304 ± 62,1277 ± 87, 4.1 ± 0.2, 6.3 ± 0.1, respectively [22].

Trans-cinnamyl acetate was found to be the major compound in cinnamon fruits, flowers, and fruit stalks. These volatile oils were found to exhibit antioxidant, antimicrobial, and antidiabetic activities. Ceylon cinnamon bark and fruits contained proanthocyanins with doubly linked bis-flavan-3-ol units in the molecule [20]. The major important difference between true cinnamon and the Cassia cinnamon has been their Coumarin content [23] Coumarins are naturally occurring plant compounds with strong anticoagulant properties which are toxic to the liver. The Coumarin content in Ceylon cinnamon appears to be none or in trace amounts [24].

Key research questions of this study were; how do consumers recognize the type of cinnamon they consume based on label attributes of the retail packs of the cinnamon value-added products? Will existing labels provide sufficient information to the global consumers and catering to their demands?

Food label, served as a communication tool between manufacturer and consumers. Consumer's ability to understand the label content and responsible labels with clear information, secure and protect the consumers. Food labels facilitate consumers to make correct food selection as well as prevent from foodborne illnesses and allergic reactions. Incorrect and misleading food labelling is a major source of frustration for buyers. Cinnamon, a plant which grows in one part of the world and consumes in another essentially needs to have responsible labelling procedures. In finding answers to the above questions, the present study aimed at assessing the clean claims of the labels of cinnamon value-added retail packs and an eyeing to identify the current level of marketing information available to the consumer. Claims were investigated through the label attributes with the main focus on the identification of the proportion of cinnamon products carrying the claims; cleaner, healthier, nutritional content, sustainability, country of origin, etc. and finding out the consumer trends of the key markets.

## Materials and methods

The present study adopted a mixed-method approach to investigate the clean label claims of cinnamon products, employing two main strategies (Fig 3). First, a survey strategy became instrumental in collecting the photographs of the value-added cinnamon retail packs available in retail chains of the USA, UK, Mexico, Japan and products of Ceylon cinnamon exporters.

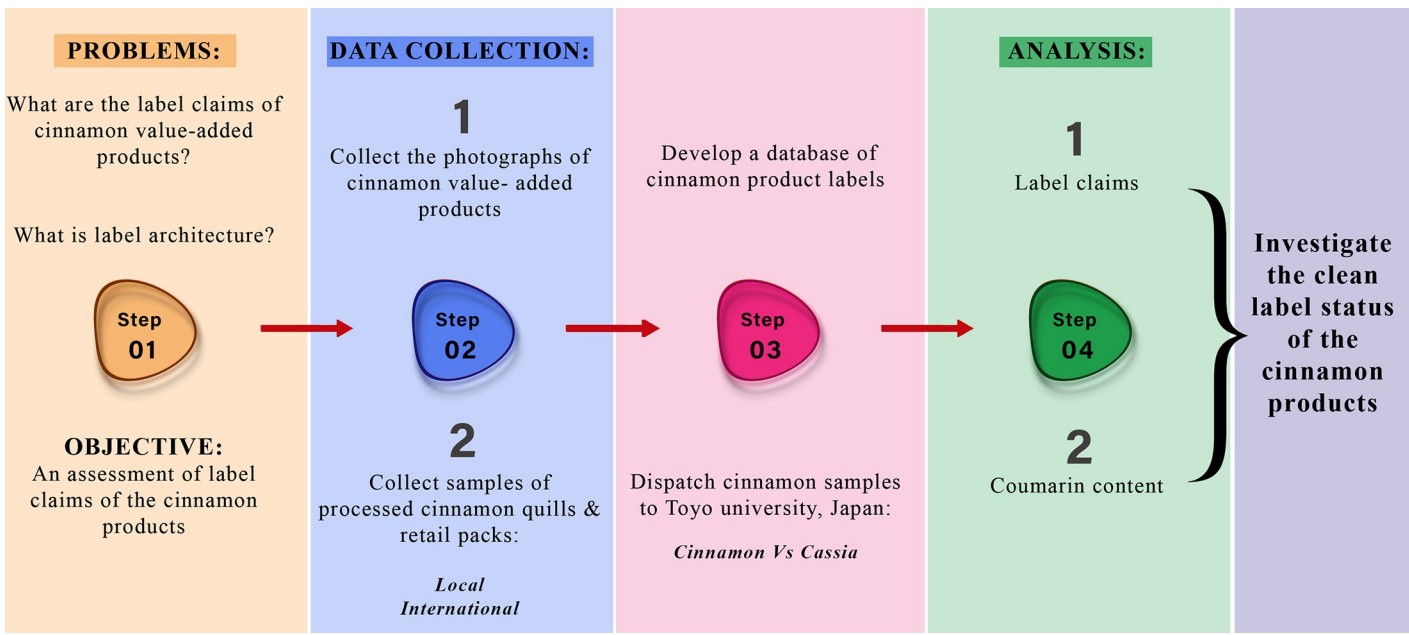

**Fig 3. Process of the research.**

The sample profile was composed of retail pack labels of cinnamon quills (48), cinnamon powder (46), cinnamon tea (54), breakfast cereals (41), bakery and confectionery (56) and nutraceuticals (31). Photographs of each product were taken; front of the pack and back of the pack, and a digital database was developed. Value-added cinnamon retail packs were categorized into 6 groups (quills, powder, tea, breakfast cereals, confectionery and bakery and nutraceuticals) for analytical purposes.

Second, processed cinnamon quills were collected from main cinnamon growing areas of Sri Lanka, and samples were dispatched for testing and experiments were conducted in Toyo University, Japan. Commercial samples were also analyzed for the comparison (Gaban, Malasiya; Hachi, Vietnam). Coumarin was analyzed according to a modified method described by the [25]. Each sample was prepared by grinding sea sand with 500 mg of cinnamon pre-treated extracts and prepared the methanol extract (with 100 ppm 4-methylumbelliferone as an internal standard). The aliquots of the solution were sonicated for 15 min and placed in a dark place for 12 h at 50°C. The aliquots were centrifuged at 20,000 g for 5 min. Liquid chromatography-tandem mass spectrometry was followed using the column of Waters Acquity HSS T3 C18 (2.1 mm×100 mm).

A content market analysis was performed to investigate the claims of the cinnamon value-added products. Mandatory and volunteer labelling requirements of Codex Alimentarius [26] were used as a guide for the label analysis. General standards for spices have not yet been prepared and they are still in discussion stage in different forums of Codex Alimentarius commission. Therefore, labelling procedure for the cinnamon products should be adapted regulations following general standards; the revised document in 2018 of the General standards for the labelling of prepackaged foods CXS 1–1985 Codex Alimentarius commission and amended in EU Regulation No. 1169/2011 of the European Parliament and of the council on the provision of food information to consumers [25, 27]. Key dimensions considered were; product name, brand, common name and botanical name of cinnamon species, nutrient profile, ingredients, product origin, label claims (clean, healthy and other), mandatory and voluntary standards and certification and consumer warnings.

Claims of clean, nutrition and other aspects which were found on cinnamon value-added retail packs were extracted, classified and added to the database [28]. Further, claims were technically validated against Codex Alimentarius (revised in 2018) and food labelling regulations of each market [20]. Label claims were investigated through label attributes. Key considerations were on identifying the proportion of cinnamon products carrying any of the claims, and the types and prevalence of claims of clean, nutrient profile, health, sustainability, nature, and country of origin, etc. Following definitions were considered as the baseline for label claims;

i. Clean claims: short, simple ingredient list, replace artificial colours, flavours, clear packing and implementing suitable practices to ensure sustainability status [29].

ii. Nutrient claims: nutrient content, amount, presence or absence of total fat, trans fat, saturated fat, cholesterol, vitamin and nutrients, sugars, sodium, poly and unsaturated fatty acids, fibre, protein and energy [30].

iii. Specific health claims of cinnamon; mainly the statements on health effects of cinnamon such as disease risk reduction claims (lowering blood sugar, cholesterol, anti-inflammable, anti-cancer, etc.), functional claims (similar effects as with Insulin hormone, in managing blood sugar levels) and consumer warnings (Coumarin content, excessive consumption, side effects, allergies, etc.). Health claims on cinnamon labels which were statements about the health effects or functional properties, which may significantly contribute to managing health status [29]. General health claims formed another set of information available for consumer and, front of the pack claims were identified and recorded by the research team based on Codex-Alimentarius and market regulations.

Moreover, the study aimed to measure the information embedded in photos of cinnamon value-added products where size and entropy related measures were considered. Photos of the same size and quality were analyzed to identify the main objects in labels of 6 groups of cinnamon value-added products. Relative size (height and width ratio of the product image), of the cinnamon quill or cinnamon name, were measured to identify the positioning of cinnamon as a promotional strategy in value-added products. Each label was weighted based on the relative image size of the cinnamon quill or name and weights ranged from 3 for equal size, 2 for half the size and 1 for a quarter of the size and 0 for non-availability of an image. Positioning strategy brings a message to the consumer, where the relative size of the term, "cinnamon", ultimately indicates the importance of the main ingredient as well as which product labels away from the strategy. According to the size of the term, "cinnamon", as positioning strategy in the label, the scores allocated were 3 for same or equal size to the other letters of product, 2 for half of the size, 1 for a quarter the size, and 0 for none availability of the term. Moreover, researchers obtained photographs of the labels of the retail packs of cinnamon value-added products exported by the Sri Lankan exporters. Export-oriented value-added products of 48 regular cinnamon exporters were considered for the analysis. Label architecture of 6 cinnamon value-added products available in 5 markets were compared with food label regulations of each market and general food label regulations of FAO. The association between existing label content and expected label attributes by regulation was tested by performing Chi-square tests. The null hypothesis; no significant association between the existing label vs the expected label attributes by regulation and alternate hypothesis for positive significant association of it were tested to find out the label status. Odds ratio calculations help to identify the mean difference of the label attributes from the regulations.

Measuring consumer perception on labels of cinnamon value-added products on purchase intentions were measured through representative focus group discussions conducted in each

**Table 1. Consumer profile of the focus group discussions.**

| Focus group | Female consumers | Male consumers | Age group | Occupation | Responsibility on food purchase (%) |
|---|---|---|---|---|---|
| USA | 6 | 4 | 31–40 = 18 | Junior executive = 10 | 70 |
| Mexico | 8 | 2 | 41–50 = 10 | Senior executive = 12 | 80 |
| UK & EU | 7 | 3 | 51–60 = 10 | Junior academic = 18 | 100 |
| Japan | 6 | 4 | >60 = 12 | Senior academic = 10 | 80 |

market destination. The study was carried out in accordance with the recommendations of the Ethical Review Committee of Sabaragamuwa University of Sri Lanka, national regulations and institutional guidelines did not require approval from an ethics committee for this research. Study based on product labels and limited number of consumers as focus group participants. All the focus group participants gave oral informed consent for anonymous use of the data for research purposes. Participants for focus groups were selected through a questionnaire and recruitment criteria were regular consumers of cinnamon value-added products, purchase from retail chains and use label information before purchase. Five focus group discussions were completed with 50 participants (Table 1). Focus groups were conducted within the university premises and the discussion formats were prepared and managed by the researchers. Retail packs of cinnamon value-added products available in each market were used for measurements. Each retail pack was evaluated individually. At the end of focus group discussions, consumers ranked the packages based on the clean status of the label.

## Results and discussion

Ceylon cinnamon and its value-added products were considered as an important natural additives and preservatives and the claim dates back to the time of Pharaoh. Trade competitiveness, the emergence of substitutes (Cassia, Kerinchi and Saigon) and value addition in a third country, have given rise to questions on cleanliness of value-added retail packs of cinnamon. Health and other claims as well as consumer warnings on excessive consumption of Coumarin, has attracted the attention of global consumers eyes on the clean status of the label and product authenticity. The experimental results show none or trace amounts of Coumarin available in Ceylon cinnamon irrespective of the agro-ecological zones they come from (Table 2). Blueprint of the clean status of cinnamon value-added retail packs available in USA, EU, UK, Japan and Mexico, reveals the present status of the label attributes (Fig 4). Cinnamon is graded by the relevant national standard of the country of production; ISO standard 6538–1997 for Cassia and 6539–2014 Ceylon cinnamon. The general food law highlighted that to guarantee food

**Table 2. Coumarin content of cinnamon.**

| Type of the Sample | Origin | Coumarin content (mg/g DW) |
|---|---|---|
| **Ceylon cinnamon sticks** | Sri Lanka (Uragaha) | $0.003 \pm 0.001^{a}$ [*] |
| **Ceylon cinnamon sticks** | Sri Lanka (Meetiyagoda) | $0.023 \pm 0.004^{a}$ |
| **Ceylon cinnamon sticks** | Sri Lanka (Ahugalla) | $0.004 \pm 0.001^{a}$ |
| **Ceylon cinnamon sticks** | Sri Lanka, Products of Gaban Co. Malaysia | $0.006 \pm 0.001^{a}$ |
| **Ceylon cinnamon powder** | Sri Lanka, Products of Gaban Co. Malaysia | $0.968 \pm 0.632^{bc}$ |
| **Cinnamon powder** | Hachi Co., Vietnam | $2.231 \pm 0.459^{b}$ |

Lower limit of detection = 0.003 mg/g DW; Lower limit of quantification 0.008 mg/g DW.

[*] Average mean values (SD) and means within a column with different superscripts are significantly different by Duncan's multiple range test (p < 0.05).

**Fig 4. Blueprint of the clean labels.** An overview of the types of claims displayed on the labels of cinnamon value added retail packs. Organic, natural, clear information, list of ingredients and traceability are sub claims of clean label claims. Nutrition claims mainly appeared as nutrient content, specific and general health claims. Country of origin, manufacturer, brand, allergens, irradiation free, non GMO, method of processing, vegetarian were other common claims. Symbols of standards and certifications appear as evidence for the consumers.

safety and to allow appropriate action in cases of unsafe food, food products must be traceable throughout the supply chain or value chain and risks of contamination must be limited. Unfortunately, value-added cinnamon products enter into the market with unknown country of origin and limited traceability.

Front of pack, back and side labels of the retail packs of cinnamon quills available in four main markets and products of Sri Lankan exporters were used only the common name, "cinnamon" and its ingredient profile. Country of origin, available standards, certifications and consumer warnings appeared as secondary level label claims (Fig 5). Similar patterns were observed in the labels of cinnamon powder products and where special claims, consumer warnings and claims on Ceylon cinnamon were considered as an important feature. Ceylon cinnamon quills carry its own identity and unique hand-crafted quill making tradition of Sri Lankans, giving an identical shape and colour which separate it from other cinnamon types. Ceylon cinnamon is considered superior to other Cassia cinnamon types due to its ultra-low levels of Coumarin (toxic, fragrant, organic chemical compound), (<0.01 mg/g DW) and existing regulations highlight the safe levels of average daily consumption of coumarin; the European Food Safety Authority (EFSA) has recommended a Coumarin daily intake limit of 0–0.1 mg/kg body weight per day [31]. According to our experimental results, one teaspoon (out of 5g) of powdered Cassia bark contains 4.85 mg of Coumarin from the samples sourced from Malaysia and 11.15 mg from the samples of Vietnam, respectively (Table 2). German Federal Institute for Risk Assessment evidenced, 1 kg of Cassia powder contains approximately 2.1–4.4 g of Coumarin [32]. This is beyond the Tolerable Daily Intake for Coumarin of 0.1mg/

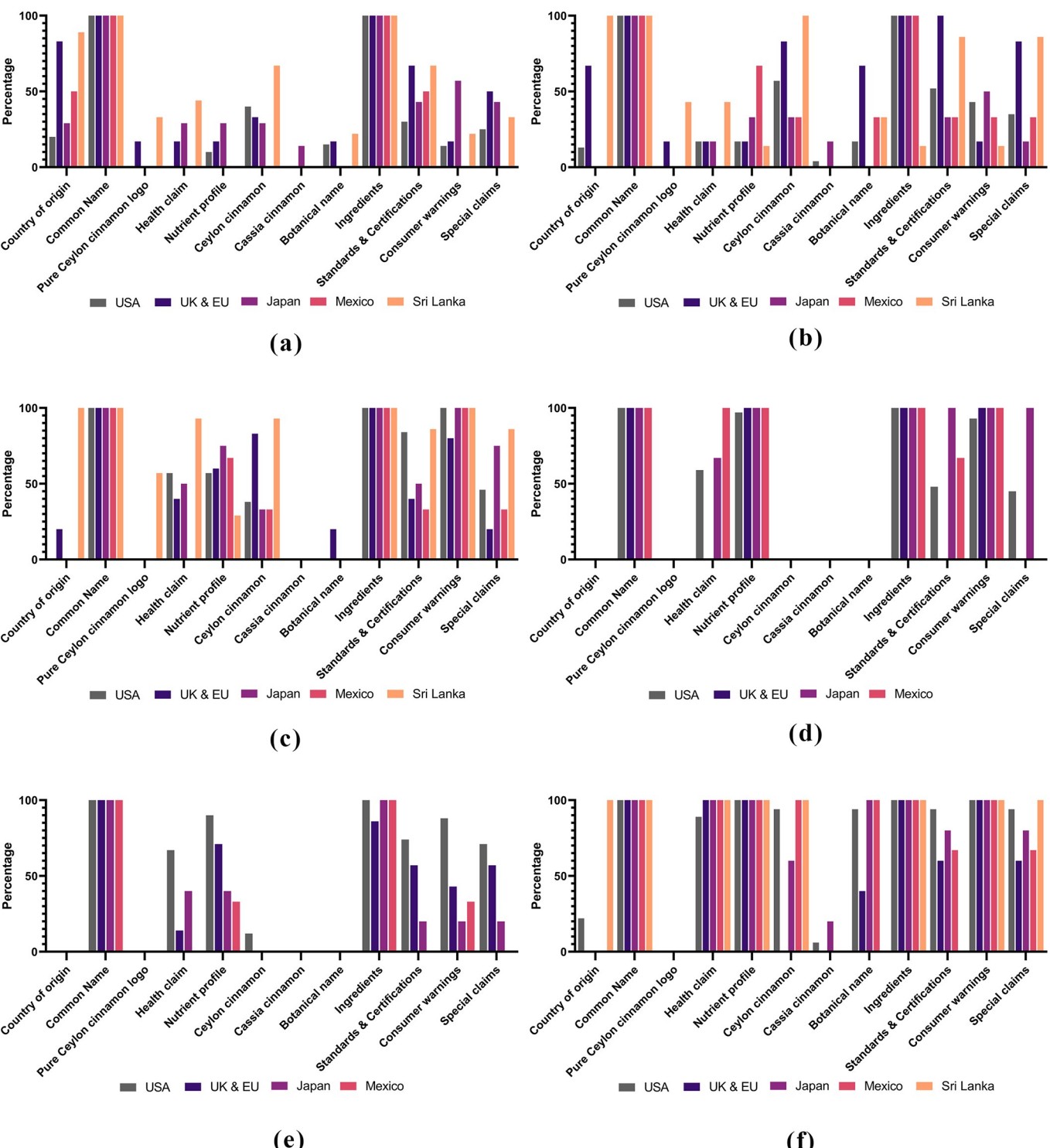

**Fig 5. Cinnamon label claims in deferent markets.** (a) Label claims of the cinnamon quills available in the main markets. Null hypothesis was rejected (Decision rule p≤ 0.05); there was a significant association between markets and label claims of cinnamon quills. Common name, pure Ceylon cinnamon logo, health claims, nutrient profile, cassia cinnamon, botanical name, and ingredients showed significant relationship. Label attributes; Country of origin (p = 0.465), Common Name (p = 0.025), Pure Ceylon cinnamon logo (p = 0.039), Health claim (p = 0.042), Nutrient profile (p = 0.042), Ceylon cinnamon (p = 0.176), Cassia cinnamon (p = 0.034), Botanical name (p = 0.042), Ingredients (p = 0.025), Standards & Certifications (p = 1.000), Consumer warnings (p = 0.080), and Special claims (p = 0.068) were considered for the hypothesis testing. (b) Label claims of the cinnamon powder available in the main markets. Null hypothesis was rejected (Decision rule p≤ 0.05); there was a significant association between markets and label claims of cinnamon powder. Common name, pure Ceylon

cinnamon logo, health claims, cassia cinnamon, and botanical name showed significant relationship. Label attributes; Country of origin (p = 0.492), Common Name (p = 0.025), Pure Ceylon cinnamon logo (p = 0.039), Health claim (p = 0.039), Nutrient profile (p = 0.102), Ceylon cinnamon (p = 0.498), Cassia cinnamon (p = 0.039), Botanical name (p = 0.031), Ingredients (p = 0.066), Standards & Certifications (p = 0.498), Consumer warnings (p = 0.068), and Special claims (p = 0.786) were considered for the hypothesis testing. (c) Label claims of the cinnamon tea available in the main markets. Null hypothesis was rejected (Decision rule p≤ 0.05); there was a significant association between markets on label claims of cinnamon tea. Common name, pure, cassia cinnamon, botanical name, ingredients and consumer warnings showed significant relationship. Label attributes; Country of origin (p = 0.258), Common Name (p = 0.025), Pure Ceylon cinnamon logo (p = 0.066), Health claim (p = 0.715), Nutrient profile (p = 0.345), Ceylon cinnamon (p = 0.684), Cassia cinnamon (p = 0.025), Botanical name (p = 0.034), Ingredients (p = 0.025), Standards & Certifications (p = 0.465), Consumer warnings (p = 0.034), and Special claims (p = 0.893) were considered for the hypothesis testing. (d) Label claims of the cinnamon based breakfast cereals available in the main markets. Null hypothesis was rejected (Decision rule p ≤ 0.05); there was a significant association between markets on label claims of cinnamon based breakfast cereals. Country of origin, common name, pure Ceylon cinnamon logo, Ceylon cinnamon, cassia cinnamon, botanical name, and ingredients showed significant relationship. Label attributes; Country of origin (p = 0.046), Common Name (p = 0.046), Pure Ceylon cinnamon logo (p = 0.046), Health claim (p = 0.581), Nutrient profile (p = 0.059), Ceylon cinnamon (p = 0.046), Cassia cinnamon (p = 0.046), Botanical name (p = 0.046), Ingredients (p = 0.046), Standards & Certifications (p = 0.854), Consumer warnings (p = 0.059), and Special claims (p = 0.450) were considered for the hypothesis testing. (e) Label claims of cinnamon based bakery & confectionery products available in the main markets. Null hypothesis was rejected (Decision rule p≤ 0.05); there was a significant association between markets on label claims of cinnamon based bakery and confectionery products. Country of origin, common name, pure Ceylon cinnamon logo, cassia cinnamon, and botanical name showed significant relationship. Label attributes; Country of origin (p = 0.046), Common Name (p = 0.046), Pure Ceylon cinnamon logo (p = 0.046), Health claim (p = 0.276), Nutrient profile (p = 0.465), Ceylon cinnamon (P = 0.059), Cassia cinnamon (p = 0.046), Botanical name (p = 0.046), Ingredients (p = 0.059), Standards & Certifications (p = 0.465), Consumer warnings (p = 0.715), and Special claims (p = 0.465) were considered for the hypothesis testing. (f) Label claims of the nutraceuticals available in the main markets. Null hypothesis was rejected (Decision rule p≤ 0.05); there was a significant association between markets on label claims of cinnamon nutraceutical. Common name, pure Ceylon cinnamon logo, health claims, nutrient profile, cassia cinnamon, ingredients, consumer warnings, and special claims showed significant relationship. Label attributes; Country of origin (p = 0.258), Common Name (p = 0.025), Pure Ceylon cinnamon logo (p = 0.025), Health claim (p = 0.034), Nutrient profile (p = 0.025), Ceylon cinnamon (p = 0.336), Cassia cinnamon (p = 0.039), Botanical name (p = 0.492), Ingredients (p = 0.025), Standards & Certifications (p = 0.500), Consumer warnings (p = 0.025), and Special claims (p = 0.043) were considered for the hypothesis testing.

kg body weight/day recommended by the EFSA. However, both types appear in the marketplace as a common term, cinnamon. Coumarin, a flavouring which can cause hepatotoxicity and the tolerable daily intake may be exceeded in consumers with high intake of cinnamon containing high levels of Coumarin [33, 34]. Ceylon (*C. zeylanicum*) and Cassia cinnamon (*C. aromaticum*) are two main types of cinnamon traded globally. Cassia cinnamon, which currently is most frequently used in food products trading globally, contains more Coumarin than lesser used Ceylon cinnamon [33].

Cinnamon quill and powder labels have ignored the most important messages on their labels and existing labels were devoid of clean claims. Label contents of cinnamon value-added products that were considered for analysis behave differently and brought a variety of information to the notice of consumer. Consumer warnings, standards and certifications, health and special claims, and mentioning of product as Ceylon cinnamon were prominent in label claims of cinnamon tea, breakfast cereals, confectionery and bakery and nutraceuticals (Fig 6). Labels of nutraceuticals were more informative and comprehensive compared to other value-added forms and manufacturers had designed the products to meet the specific regulatory requirements of the markets. Nutraceuticals and labels of cinnamon value-added products available in EU and USA markets were closer to the clean claims whereas cinnamon quills, powder and products available in Mexican market were found to be far away from the clean concepts with very limited information.

Hypothesis testing showed the association between present label attributes and expected label attributes of 6 different cinnamon value-added products available in 5 different markets (Table 3). Cinnamon quills available in all marketplaces show no association between the attributes of present labels and expected labels by the food law. In contrast, labels of cinnamon powder in the USA market as well as the Sri Lankan exporters showed a positive association between the attributes of present labels and expected labels. Furthermore, cinnamon tea labels followed a similar pattern and both cinnamon tea labels found in the USA market as well as the cinnamon tea exports from Sri Lanka were in line with the expected labeling requirements. Moreover, cinnamon breakfast cereals, confectionery and bakery products, and nutraceuticals

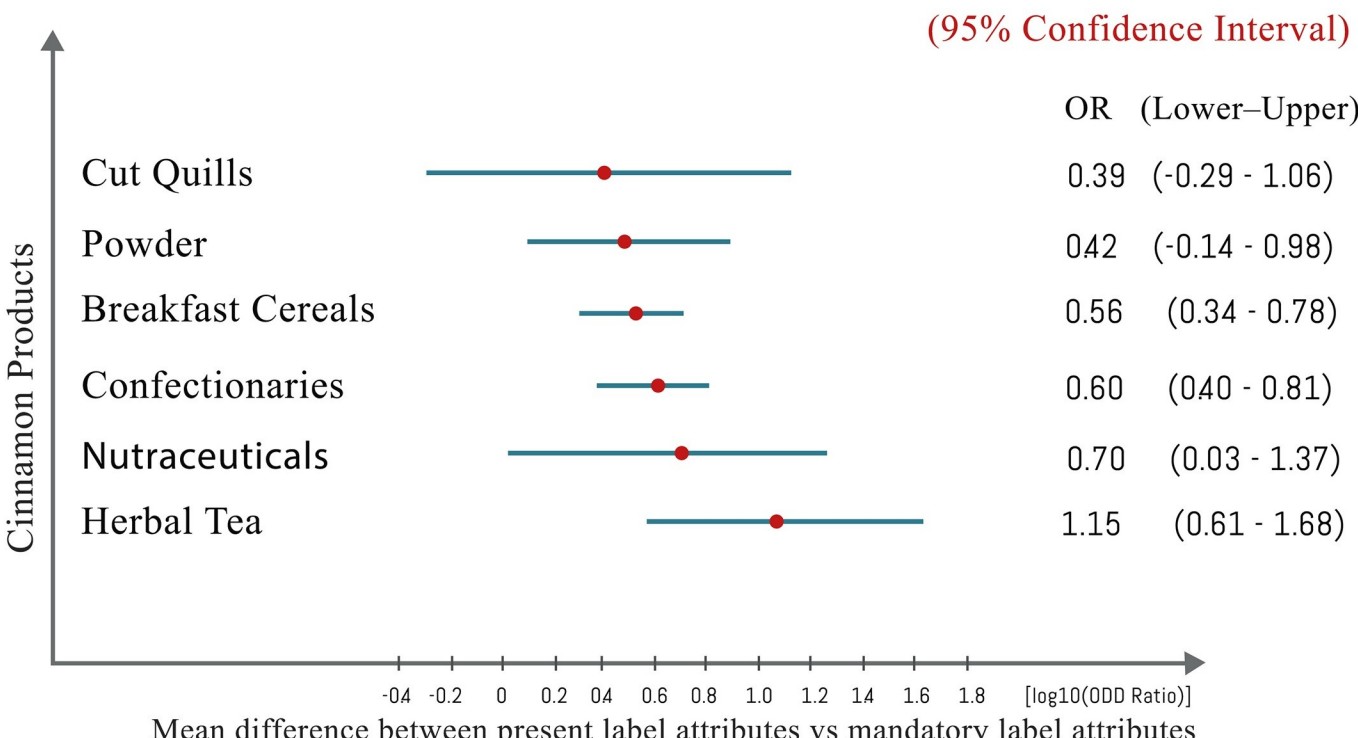

**Fig 6. Mean difference of the present label attributes with labelling requirements of FAO (Codex Alimentarius).** The no. of observations in each category was cinnamon quills (48), cinnamon powder (46), breakfast cereals (41), Confectionery and bakery (56), Nutraceuticals (31), and cinnamon tea (54). Mean label attribute sizes shown with 95% confidence interval.

of the USA market have positive association between current label attributes vs expected labels. Label attributes of cinnamon value-added products for all other markets considered were in need to improve their status to meet the regulatory expectations. Odds ratio analysis derived the holistic view on present label attributes of 6 different cinnamon products were considered for 5 marketplaces.

Size and entropy related measures of label analysis aimed to identify the product positioning strategy based on cinnamon as the main ingredient. Relative size of the image (cinnamon quill), type of cinnamon (true/Ceylon cinnamon or Cassia cinnamon) and the term,

**Table 3. Association between present label attributes vs expected label attributes of 6 different cinnamon value-added products.**

| Countries | | Quills | Powder | Tea | Cereals | Confectionaries & Bakery | Nutraceuticals |
|---|---|---|---|---|---|---|---|
| USA | P value | 0.133 | 0.288 | 0.873 | 0.470 | 0.781 | 0.067 |
| | Confidence interval | -60.0 (-85.0,14.0) | -31.0 (-70.0, 17.0) | 0.00 (-54.0, 57.0) | -3.0 (-100, 48.0) | 0.00 (-76.0, 67.0) | 88.0 (0.00, 94.0) |
| Europe & UK | P value | 0.586 | 0.604 | 0.843 | 0.248 | 0.202 | 0.680 |
| | Confidence interval | -16.0 (-66.0, 33.0) | 17.0 (-66.0, 67.0) | 0.00 (-60.0, 40.0) | 0.00 (-100, 0.00) | -29.0 (-100, 14.0) | 0.00 (-40.0, 60.0) |
| Japan | P value | 0.205 | 0.176 | 0.874 | 0.731 | 0.092 | 0.123 |
| | Confidence interval | -28.0 (-57.0, 29.0) | -50.0 (-83.0, 17.0) | 0.00 (-50.0, 50.0) | 0.00 (-33.0, 100) | -60.00 (-100, 0.00) | 60.0 (0.00, 100) |
| Mexico | P value | 0.058 | 0.183 | 0.370 | 0.796 | 0.044 | 0.166 |
| | Confidence interval | -50.0 (-100, 0.00) | -34.0 (-67.0, 33.0) | -33.0 (-100.0, 33.0) | 0.00 (-100, 67.0) | -67.00 (-100, 0.00) | 67.0 (0.00, 100) |
| Sri Lanka | P value | 0.812 | 0.726 | 0.152 | - | - | 0.248 |
| | Confidence interval | 0.00 (-56.0, 34.0) | 7.0 (-53.0, 72.0) | 57.0 (-7.0, 93.0) | - | - | 0.00 (0.00, 100) |

"cinnamon", were analyzed in all product categories across the main markets. In general, in respect of the types of cinnamon, images were available of Cassia cinnamon but none from true cinnamon. Low level of consumer awareness in the international market on the difference between true cinnamon and Cassia cinnamon allows manufacturers to provide misleading label information. Present labelling behaviour was found to be far away from clean concepts and claims but more closer to dirty labels. On the other hand, labels of the Sri Lankan exporters who were considered, displayed the true cinnamon quill for its authenticity, but their limited market operations tend to hinder the information available for consumers.

Analysis of label objects, images and words, facilitate to understand the clean claims of the labels via its product positioning strategy; cinnamon as a key ingredient. Cinnamon based value-added forms; breakfast cereals, cinnamon tea, confectionery and bakery and nutraceuticals, utilized various positioning strategies, objects and words to position products in the marketplace. Fig 7 explains the present status of the label objects in the main markets. Equal size letters have been used for the name of the product and the term "cinnamon" in 4 main markets while letters in half the size height use for all markets. Positioning approach caters to the information necessitated by the consumers. On the other hand, half the size image of cinnamon was found in the packages of breakfast cereals of 3 main markets but all pictures were of Cassia cinnamon. None of the labels used pictures of true cinnamon and which may indirectly indicate the type of ingredient or cinnamon available in breakfast cereals. In contrast, label object analysis of cinnamon tea showed positive clean claims, where equal and half the size of height of term, "cinnamon" and an image of cinnamon appeared in all labels. Labels in cinnamon tea were cleaner compared to breakfast cereals (Fig 7). Unfortunately, none of the labels displayed any image of true cinnamon, except in cinnamon tea exported from Sri Lanka.

Cinnamon is widely used globally in bakery and confectionery industry, as a key ingredient. Labels of bakery and confectionery products used for analysis showed a less clean status compared to that of breakfast cereals and tea (Fig 8). A limited number of products have displayed the term "cinnamon" in equal and half the size in front of the pack while showing an image of cinnamon. In contrast, label objects of nutraceuticals displayed both the term and an image of

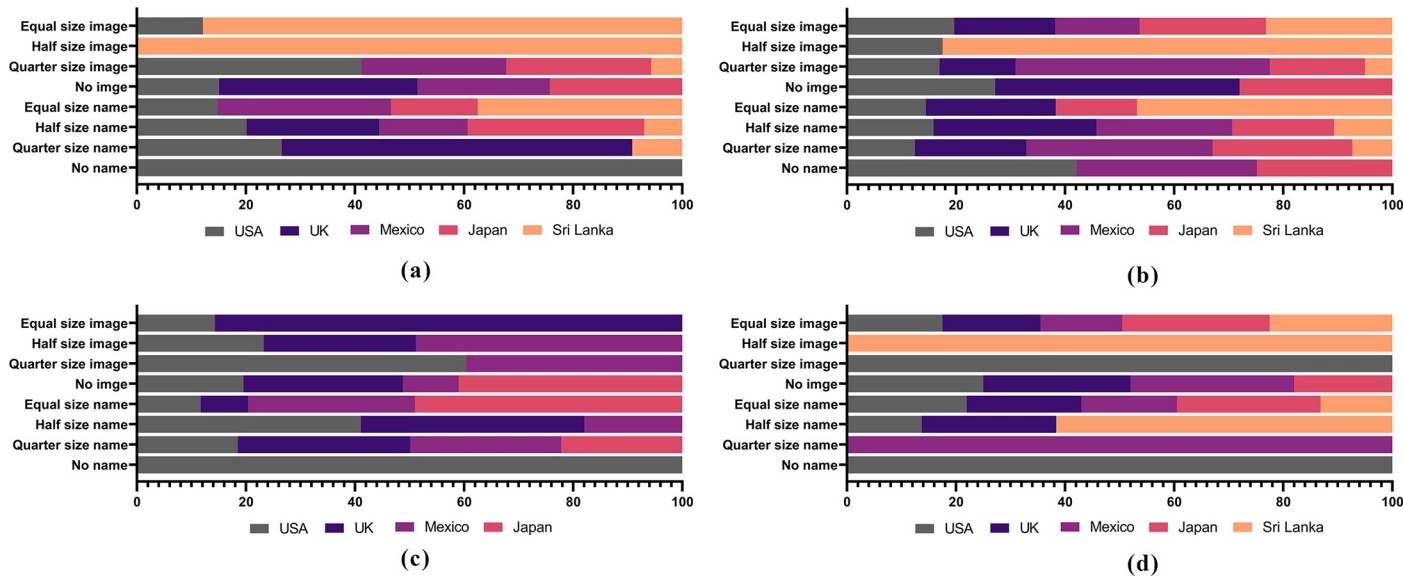

**Fig 7. Assessment of positioning strategy through label objects.** Assessment of positioning strategy through label objects: an image of cinnamon and word, cinnamon in cinnamon-based breakfast cereals (a); cinnamon tea (b); cinnamon based bakery and confectionery (c); and nutraceuticals (d).

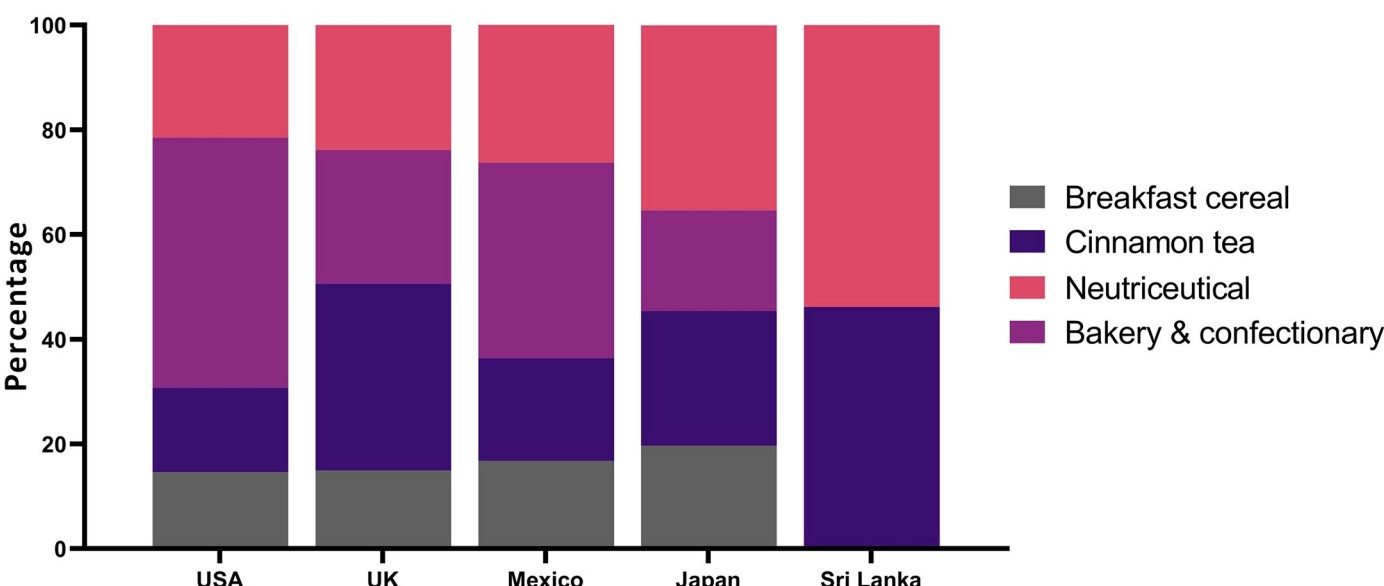

**Fig 8. Clean label status of the cinnamon value-added retail products in the main markets.** Bakery and confectionery of USA market, cinnamon tea of UK/EU market, bakery and confectionery of Mexican market, Nutraceuticals of Japanese and Sri Lankan exporters were recognized as the market with key clean labeled products.

"cinnamon" in equal size (Fig 8). None of the labels have shown the image of true cinnamon in labels and declaration of this information is mandatory for all markets considered. Label contents were measured and scores were allocated to rank the clean status of labels in each main market. Best clean claims were recorded for bakery and confectionery category in the USA market, breakfast cereals in the Japanese market, cinnamon tea and nutraceuticals from the Sri Lankan exporters.

Focus group discussions were used to sketch the characteristics of the global cinnamon consumer. Consumer base was composed of the young and mid-aged (68%) and the very elderly (24%). Occupational profile represented 56% of junior executives and academics with postgraduate qualifications. The majority (65%) of them were responsible for the purchase of food for their households. All selected consumers were regular consumers of cinnamon as breakfast cereal, bakery products, confectionery, etc. In general, an average consumer had no clear idea on the difference between the true cinnamon and Cassia cinnamon. The purpose of their consumption of cinnamon lay in the taste and aroma (85%) of cinnamon, and its health claims (15%). A limited number of consumers (12%) were able to recognize the difference between true cinnamon and Cassia and the health impacts of true cinnamon. On the other hand, clean labels; brief ingredient list, chemical composition especially the Coumarin and phytochemical properties, simple and clear packaging, geographical indication and traceability were found to be demanded by the cinnamon consumers. True or Ceylon cinnamon's share of the world market is about 8% and the lion's share is held by the Cassia cinnamon. High level of Coumarin in cassia and its impact on human health was recognized and published as a scientific truth [34]. confirmed that high levels of Coumarin in Cassia on quantification of flavoring constituents in cinnamon based on German retail market. Risk assessment of Coumarin intake of Norwegian nationals [31] also confirmed the possible adverse health risks of Coumarin consumption by exceeding tolerable daily intake 3-fold for 1–2 times a week. Further, a case of famous Danish cinnamon roll, Danish Food Safety Authority has shown that the product significantly exceeds the limit for daily intake [35]. Clean status of the labels, especially, which

type of cinnamon was used in value addition; true cinnamon or Cassia forms an essential piece of information that should be shared with consumers.

## Conclusions

For centuries, Ceylon cinnamon is a natural flavouring additive and a preservative that has been providing numerous health benefits to the consumer through the food they consume. *C. zeylanicum* has a health favourable chemical profile and does not contain Coumarin, a carcinogenic compound, which is present in the other most consumable cinnamon types in the global market. It is an obvious fact that the consumer has the right to know what they consume and in which quantities. The only possible way of signaling or providing such information is the package and its label. Key label claims of cinnamon products consist of traits such as clean, health, nutrient and sustainability. Cinnamon quill and powder, which are basic value-added forms, have ignored the regulatory and mandatory labeling requirements of the main markets. In contrast, labels of breakfast cereals, bakery and confectionery moderately cater to the demands of clean status. Cinnamon tea and nutraceutical labels were closer to clean status by providing mandatory information requirements of consumers. Unfortunately, about 90% of labels that were considered by this research, displayed only the images of Cassia cinnamon, apart from the labels of Ceylon cinnamon exporters. Clean labels carrying mandatory labelling information formed an asset to consumers of USA, UK, Japan and buyers of Sri Lankan cinnamon exporters. As the main cinnamon buyer and largest traditional consumer, Mexico was far behind the mandatory labelling requirements and clean status. Cinnamon tea and nutraceuticals available in the USA market were able to claim the clean label status. Cinnamon consumers were concerned about clean labels; brief ingredient list, chemical composition, especially the Coumarin and phytochemical status, simple and clear packaging, geographical indication and traceability.

As there are different cinnamon varieties with different chemical contents, which differ each other by their chemical properties, available in the market, the authors of this paper suggest to include the botanical name (*Linnaeus nomenclature*) of the cinnamon species, from which derived the product to the label in order not to mislead or confuse the consumer.

## Supporting information

**S1 Dataset. Coumarin content of cinnamon, coumarin content of six deferent samples were analyzed in order to compare the constituent.**
(XLSX)

**S2 Dataset. Cinnamon label claims in deferent markets, label claims as country of origin, common name, pure Ceylon cinnamon logo, health claim, nutrient profile Ceylon cinnamon, Cassia cinnamon, botanical name, ingredients, standards and certifications, consumer warnings and Special claims were considered for six different products.**
(XLSX)

**S3 Dataset. Positioning strategy through label objects, sizes of the cinnamon image and wording appeared in the packaging considered for the assessment of cinnamon based breakfast cereals; cinnamon tea; cinnamon based bakery and confectionery; and nutraceutical.**
(XLSX)

**S4 Dataset. Clean label status of the cinnamon value-added retail products in the main markets, bakery and confectionery of USA market, cinnamon tea of UK/EU market, bakery and confectionery of Mexican market, Nutraceuticals of Japanese and Sri Lankan exporters**

**were recognized as the market with key clean labeled products.**
(XLSX)

**S5 Dataset. Disclose of brands, products in different regions were identified with categories and brand names to evaluated the label content.**
(XLSX)

## Acknowledgments

We are grateful to Mr. Sarada De Silva for his dedicated and expeditious help by arranging field work as well as in linking up the research team with overseas markets. We like to thank the participants in our focus groups for their time and willingness to share their experiences with us. We thank the project coordinators of the special project on cinnamon, Prof. Ranjith Senarathne, Prof. K.D.N. Weerasinghe and our contact persons at the National Science Foundation for their enormous efforts in the development of cinnamon research. Our special thanks go to Prof. Oscar Amarasinghe and Dr. Mahesh De Silva for their constructive comments on finalizing the manuscript.

## Author Contributions

**Conceptualization:** Devarahandhi Achini Melda De Silva, Rajapakshage Heshani Navoda Rajapaksha.

**Data curation:** Renda Kankanamge Chaturika Jeewanthi, Rajapakshage Heshani Navoda Rajapaksha, Weddagala Mudiyanselage Tharaka Bilindu Weddagala, Naoki Hirotsu, Bun-ichi Shimizu.

**Formal analysis:** Renda Kankanamge Chaturika Jeewanthi, Naoki Hirotsu.

**Funding acquisition:** Devarahandhi Achini Melda De Silva.

**Investigation:** Devarahandhi Achini Melda De Silva, Renda Kankanamge Chaturika Jeewanthi, Naoki Hirotsu, Bun-ichi Shimizu, Munasinghe Arachchige Jagath Priyantha Munasinghe.

**Methodology:** Renda Kankanamge Chaturika Jeewanthi.

**Project administration:** Devarahandhi Achini Melda De Silva.

**Resources:** Devarahandhi Achini Melda De Silva.

**Software:** Devarahandhi Achini Melda De Silva, Rajapakshage Heshani Navoda Rajapaksha, Weddagala Mudiyanselage Tharaka Bilindu Weddagala, Naoki Hirotsu.

**Supervision:** Devarahandhi Achini Melda De Silva.

**Validation:** Devarahandhi Achini Melda De Silva, Munasinghe Arachchige Jagath Priyantha Munasinghe.

**Visualization:** Naoki Hirotsu.

**Writing – original draft:** Devarahandhi Achini Melda De Silva, Renda Kankanamge Chaturika Jeewanthi, Weddagala Mudiyanselage Tharaka Bilindu Weddagala, Munasinghe Arachchige Jagath Priyantha Munasinghe.

**Writing – review & editing:** Renda Kankanamge Chaturika Jeewanthi.

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
