## [Decision Letter · Decision Letter 0]

19 Apr 2021

PONE-D-21-06322

Clean vs dirty labels: transparency and authenticity of the labels of Ceylon cinnamon

PLOS ONE

Dear Dr. De Silva,

Thank you for submitting your manuscript to PLOS ONE. After careful consideration, we feel that it has merit but does not fully meet PLOS ONE’s publication criteria as it currently stands. Therefore, we invite you to submit a revised version of the manuscript that addresses the points raised during the review process.

We look forward to receiving your revised manuscript.

Kind regards,

Christophe Hano

Academic Editor

PLOS ONE

Journal Requirements:

3. In your Methods, please provide the following information:

a) The exact sources of all samples and products used in your study.

b) Any product brand names that would enable an interested researcher to replicate the sampling.

4. For copyright reasons, we ask that you please remove the logos from Fig. 3.

Reviewers' comments:

Reviewer's Responses to Questions

**Comments to the Author**

1. Is the manuscript technically sound, and do the data support the conclusions?

Reviewer #1: Yes

Reviewer #2: Partly

2. Has the statistical analysis been performed appropriately and rigorously? 

Reviewer #1: Yes

Reviewer #2: Yes

3. Have the authors made all data underlying the findings in their manuscript fully available?

Reviewer #1: Yes

Reviewer #2: Yes

4. Is the manuscript presented in an intelligible fashion and written in standard English?

Reviewer #1: Yes

Reviewer #2: Yes

5. Review Comments to the Author

Reviewer #1: 1. This manuscript dealt with the impact of labels on the marketing of Ceylon cinnamon. The resulting data are practicable for the related field.

2. The resolution of all figures is low and thus it's very difficult to analyze the data.

3. Some grammar errors are listed as follows and they should be improved.

4. Abstract: market place - marketplace; neutriceuticals - nutriceuticals?; seventy six - seventy-six; liewithin - lie within

5. line 3: to 14th - to the 14th

7. line 11: nutritional quality and taste is - nutritional quality and taste are

8. line 9: a significant - significant

9. line 462: signalling - signaling

10. line 466: labelling - labeling

11. line 475: in USA market - in the USA market

12. line 475: were concern on - were concern about

13. The keyword "Clean label Content marketing" seems not appropriate. How about "Ingredient transparency"?

Reviewer #2: Comment and suggestion to authors:

Manuscript ID: PONE-D-21-06322

Manuscript title: Clean vs dirty labels: transparency and authenticity of the labels of Ceylon cinnamon

1) Line63-66: “…Kerinci cinnamon (Cinnamomum burmannii) by Indonesia, Cassia cinnamon (Cinnamomum cassia) by China, Saigon cinnamon (Cinnamomum loureiroi) by Vietnam, and True cinnamon or Ceylon cinnamon (Cinnamomum zelanicum) by Sri Lanka.” The authors must pay more attention to check and use the accepted name (legitimate name) of plant species.

2) “Cinnamomum zelanicum” is the misspelling one, please carefully check the whole manuscript and correct it.

3) The authors emphasized on “True cinnamon or Ceylon cinnamon”, so its clear photo as well as the morphological characters should be provided comparing with the other commercial cinnamon.

4) The Figure 1: a) and b) should be revised and presented using high resolution.

5) The Figure 4: a) to f) should be revised and presented in the way that helps the readers easy to follow, and can see the main points easily.

6) The previous published works that related to this study should be added to discuss with these results.

7) There are many spelling mistakes and grammatical error found in this manuscript, the author should pay more attention on this point and check the whole manuscript before re-submission.

6. PLOS authors have the option to publish the peer review history of their article (what does this mean?). If published, this will include your full peer review and any attached files.

Reviewer #1: No

Reviewer #2: No

---

## [Author Response · Author response to Decision Letter 0]

15 Jun 2021

Reviewers' comments:

Reviewer #1: 

1. This manuscript dealt with the impact of labels on the marketing of Ceylon cinnamon. The resulting data are practicable for the related field. 

Response: Thank you for your positive interest in this paper and for your review. 

2. The resolution of all figures is low and thus it's very difficult to analyze the data. 

Response: Thank you for your comments on figures and we have improved all figures

3. Some grammar errors are listed as follows and they should be improved. 

Response: Thank you for pointing them. We revised as your suggestion. 

4. Abstract: market place - marketplace; neutriceuticals - nutriceuticals?; seventy six - seventy-six; liewithin - lie within 

Response: Revised 

5. line 3: to 14th - to the 14th 

Response: Revised 

7. line 11: nutritional quality and taste is - nutritional quality and taste are 

Response: Revised 

8. line 9: a significant - significant 

Response: Revised 

9. line 462: signalling - signaling 

Response: Revised 

10. line 466: labelling - labeling 

Response: Revised 

11. line 475: in USA market - in the USA market 

Response: Revised 

12. line 475: were concern on - were concern about 

Response: Revised 

13. The keyword "Clean label Content marketing" seems not appropriate. How about "Ingredient transparency"?

Response: Revised as your suggestion

Reviewer #2: 

1) Line63-66: “...Kerinci cinnamon (Cinnamomum burmannii) by Indonesia, Cassia cinnamon (Cinnamomum cassia)

by China, Saigon cinnamon (Cinnamomum loureiroi) by Vietnam, and True cinnamon or Ceylon cinnamon

(Cinnamomum zelanicum) by Sri Lanka.” The authors must pay more attention to check and use the accepted name

(legitimate name) of plant species.

Response: Thank you for your comment. “Kerinci” has been changed as the Indonesian cinnamon. Kerinci is one specific region of Indonesia that Cinnamomum burmannii comes from and it is a common term when comes to Indonesian cinnamon. The other names that we mentioned here can be considered as accepted names in the industry that are using commonly.

2) “Cinnamomum zelanicum” is the misspelling one, please carefully check the whole manuscript and correct it.

Response: Revised 

3) The authors emphasized on “True cinnamon or Ceylon cinnamon”, so its clear photo as well as the morphological characters should be provided comparing with the other commercial cinnamon.

Response: We have included details of both Ceylon cinnamon and Cassia

4) The Figure 1: a) and b) should be revised and presented using high resolution.

Response: Revised 

5) The Figure 4: a) to f) should be revised and presented in the way that helps the readers easy to follow, and can see the main points easily.

Response: Revised 

6) The previous published works that related to this study should be added to discuss with these results.

Response: Thank you for your suggestion. The manuscripts’ main focus is to discuss the mislabeling of cinnamon products where it needs clear specifications to understand the difference between different types of cinnamon in the current market. Since the demarcations between these cinnamon species are having a greater impact on different health claims, we believe that the consumer of the product should have a right to be acknowledged of what they consume.

We could find the studies which were based on the differences between the cinnamon species and supported that information to build up the discussion of this manuscript. However, there is lack of original studies (reference no. 29 and 33) on this area which directly analyzed or evaluated the labeling effect of cinnamon products which imparts the consumer with the differences between cinnamon types. The studies were in reference 29 and 33 are few

7) There are many spelling mistakes and grammatical error found in this manuscript, the author should pay more attention on this point and check the whole manuscript before re-submission.

Response: Thanks for your comment towards the improvement of this manuscript. The revised places after careful observations were highlighted by red fonts. 

Editor comments:

1. Please ensure that your manuscript meets PLOS ONE’s style requirements, including those for file naming, etc.

Response: We have re-shaped the manuscript in line with PLOS ONE requirements, including the file naming, etc.

Note: Published articles of PLOS ONE carry title numbering system but we couldn’t find it in guidelines. If necessary we are happy to insert the numbers

2. We suggest you thoroughly copyedit your manuscript for language usage, spelling and grammar. 

Response: Thank you for your comments and we have obtained the professional editing support from Dr. Premachandra Wattage, School of Earth & Environmental Sciences, University of Portsmouth, UK and Town & Country Planning, University of Moratuwa, Moratuwa, Sri Lanka, (Guest Editor of the MDPI/Sustainability and Land Journals)

Further, we have incorporated a copy of manuscript showing changes in a supporting information file.

Clean copy of the edited manuscript in a separate file

3. In your methods, Please provide the following information. 

a. The exact sources of all samples and products use in your study

b. Any product bard names that would enable an interested researcher to replicate the sampling

Response: Thank you for pointing them. We have revised the methods section as you suggested and insert the sources of samples. 

ii. Retail packs of cinnamon value-added products (quills, powder, tea, breakfast cereals, confectionery and bakery and nutraceuticals) used in USA, UK, Mexico, Japan and products of Sri Lankan cinnamon exporters

b. Any product brand names that would enable an interested researcher to replicate the sampling

Response: Thank you for the suggestion. We do agree that sharing information would facilitate future research avenues. We have made arrangements to submit our data base on cinnamon value-added retail packs with brand names as separate supplementary file. Further, we are unable to declare the brand names in methodology section due to ethical reasons that may negatively affected on brands as well as the businesses. 

4. For copy right reasons, we ask that you please remove the logos from fig.3

Response: Revised the fig.3. We only keep the Ceylon cinnamon logo which we received the permission from Export Development Board of Sri Lanka, to use for research purpose.

5. Ethics statement should only appear in the methods section of your manuscript. 

 Response: Revised as recommended 

Further, we would like to make changes to the “financial disclosure” and please find the updated version below. 

“The empirical research (market survey) informing this paper was funded by the Ministry of Social Welfare and Primary Industries of Sri Lanka and the National Science Foundation. DAM received the grant for project titled, Ceylon cinnamon value chain development: Making global market space for value chain actors (SP/CIN/2016/05). DAM and RKC received the research, innovation and commercialization grant of AHEAD (Accelerating Higher Education and Development) project for the development of novel beverages from Ceylon cinnamon (6026-LK/8743-LK/ P159995) which facilitate the chemical profiling process. The funders had no role in study design, data collection and analysis, decision to publish or preparation of the manuscript.”

---

## [Decision Letter · Decision Letter 1]

13 Aug 2021

PONE-D-21-06322R1

Clean vs dirty labels: Transparency and authenticity of the labels of Ceylon cinnamon

PLOS ONE

Dear Dr. De Silva,

Thank you for submitting your manuscript to PLOS ONE. After careful consideration, we feel that it has merit but does not fully meet PLOS ONE’s publication criteria as it currently stands. Therefore, we invite you to submit a revised version of the manuscript that addresses the points raised during the review process.

We look forward to receiving your revised manuscript.

Kind regards,

Christophe Hano

Academic Editor

PLOS ONE

Journal Requirements:

Reviewers' comments:

Reviewer's Responses to Questions

**Comments to the Author**

1. If the authors have adequately addressed your comments raised in a previous round of review and you feel that this manuscript is now acceptable for publication, you may indicate that here to bypass the “Comments to the Author” section, enter your conflict of interest statement in the “Confidential to Editor” section, and submit your "Accept" recommendation.

Reviewer #1: All comments have been addressed

Reviewer #2: All comments have been addressed

2. Is the manuscript technically sound, and do the data support the conclusions?

Reviewer #1: Yes

Reviewer #2: Yes

3. Has the statistical analysis been performed appropriately and rigorously? 

Reviewer #1: Yes

Reviewer #2: Yes

4. Have the authors made all data underlying the findings in their manuscript fully available?

Reviewer #1: Yes

Reviewer #2: Yes

5. Is the manuscript presented in an intelligible fashion and written in standard English?

Reviewer #1: Yes

Reviewer #2: Yes

6. Review Comments to the Author

Reviewer #1: (No Response)

Reviewer #2: Comment and suggestion to authors:

Manuscript ID: PONE-D-21-06322R1

Titled: "Clean vs dirty labels: Transparency and authenticity of the labels of Ceylon cinnamon"

1.) The authors must pay more attention on how to write the scientific name, specific epithet must be written in lowercase. For example, “Cinnamomum Cassia” in the figure 2 is incorrect. Please, check the whole manuscript to correct this point.

2.) There are some spelling mistakes and grammatical error found in this manuscript, the author should pay more attention on this point and check the whole manuscript before re-submission.

7. PLOS authors have the option to publish the peer review history of their article (what does this mean?). If published, this will include your full peer review and any attached files.

Reviewer #1: No

Reviewer #2: No

---

## [Author Response · Author response to Decision Letter 1]

18 Aug 2021

Thank you for point out the mistake in scientific name and the corrections are made in the figure two as well as in the manuscript text.

Spelling mistakes and grammatical error found in this manuscript were corrected

---

## [Decision Letter · Decision Letter 2]

26 Sep 2021

PONE-D-21-06322R2

Clean vs dirty labels: Transparency and authenticity of the labels of Ceylon cinnamon

PLOS ONE

Dear Dr. De Silva,

Thank you for submitting your manuscript to PLOS ONE. After careful consideration, we feel that it has merit but does not fully meet PLOS ONE’s publication criteria as it currently stands. Therefore, we invite you to submit a revised version of the manuscript that addresses the points raised during the review process.

We look forward to receiving your revised manuscript.

Kind regards,

Christophe Hano

Academic Editor

PLOS ONE

Journal Requirements:

Additional Editor Comments (if provided):

Reviewers' comments:

Reviewer's Responses to Questions

**Comments to the Author**

1. If the authors have adequately addressed your comments raised in a previous round of review and you feel that this manuscript is now acceptable for publication, you may indicate that here to bypass the “Comments to the Author” section, enter your conflict of interest statement in the “Confidential to Editor” section, and submit your "Accept" recommendation.

Reviewer #1: All comments have been addressed

Reviewer #2: All comments have been addressed

2. Is the manuscript technically sound, and do the data support the conclusions?

Reviewer #1: Yes

Reviewer #2: Yes

3. Has the statistical analysis been performed appropriately and rigorously? 

Reviewer #1: Yes

Reviewer #2: Yes

4. Have the authors made all data underlying the findings in their manuscript fully available?

Reviewer #1: Yes

Reviewer #2: Yes

5. Is the manuscript presented in an intelligible fashion and written in standard English?

Reviewer #1: Yes

Reviewer #2: Yes

6. Review Comments to the Author

Reviewer #1: (No Response)

Reviewer #2: Comment and suggestion to authors:

Manuscript ID: PONE-D-21-06322R2

Titled: "Clean vs dirty labels: Transparency and authenticity of the labels of Ceylon cinnamon"

1.) After writing the full scientific name for the first time, next time, the generic name should be written as an abbreviation. For example, the authors write “Cinnamomum zeylanicum” for the first time, the next time should use C. zeylanicum. Please, check the whole manuscript to correct this point.

2.) There are some spelling mistakes and grammatical error found in this manuscript, please check and correct before re-submission.

7. PLOS authors have the option to publish the peer review history of their article (what does this mean?). If published, this will include your full peer review and any attached files.

Reviewer #1: No

Reviewer #2: No

---

## [Author Response · Author response to Decision Letter 2]

15 Oct 2021

Reviewers' comments:

Reviewer's Responses to Questions

Comments to the Author

6. Review Comments to the Author

Reviewer #1: (No Response)

Reviewer #2: Comment and suggestion to authors:

Manuscript ID: PONE-D-21-06322R2

Titled: "Clean vs dirty labels: Transparency and authenticity of the labels of Ceylon cinnamon"

1.) After writing the full scientific name for the first time, next time, the generic name should be written as an abbreviation. For example, the authors write “Cinnamomum zeylanicum” for the first time, the next time should use C. zeylanicum. Please, check the whole manuscript to correct this point.

Thank you very much for point out the error and the manuscript is corrected accordingly. 

2.) There are some spelling mistakes and grammatical error found in this manuscript, please check and correct before re-submission.

Thank you for point out the issue. We have corrected with spelling mistakes and grammatical error thoroughly.

---

## [Decision Letter · Decision Letter 3]

11 Nov 2021

Clean vs dirty labels: Transparency and authenticity of the labels of Ceylon cinnamon

PONE-D-21-06322R3

Dear Dr. De Silva,

We’re pleased to inform you that your manuscript has been judged scientifically suitable for publication and will be formally accepted for publication once it meets all outstanding technical requirements.

Kind regards,

Christophe Hano

Academic Editor

PLOS ONE

Additional Editor Comments (optional):

Reviewers' comments:

Reviewer's Responses to Questions

**Comments to the Author**

1. If the authors have adequately addressed your comments raised in a previous round of review and you feel that this manuscript is now acceptable for publication, you may indicate that here to bypass the “Comments to the Author” section, enter your conflict of interest statement in the “Confidential to Editor” section, and submit your "Accept" recommendation.

Reviewer #2: All comments have been addressed

2. Is the manuscript technically sound, and do the data support the conclusions?

Reviewer #2: Yes

3. Has the statistical analysis been performed appropriately and rigorously? 

Reviewer #2: Yes

4. Have the authors made all data underlying the findings in their manuscript fully available?

Reviewer #2: Yes

5. Is the manuscript presented in an intelligible fashion and written in standard English?

Reviewer #2: Yes

6. Review Comments to the Author

Reviewer #2: Manuscript ID: PONE-D-21-06322R3

Titled: "Clean vs dirty labels: Transparency and authenticity of the labels of Ceylon

cinnamon"

Accepted in this present form

7. PLOS authors have the option to publish the peer review history of their article (what does this mean?). If published, this will include your full peer review and any attached files.

Reviewer #2: No

---

## [Editor Report · Acceptance letter]

15 Nov 2021

PONE-D-21-06322R3 

Clean vs dirty labels: Transparency and authenticity of the labels of Ceylon cinnamon 

Dear Dr. De Silva:

I'm pleased to inform you that your manuscript has been deemed suitable for publication in PLOS ONE. Congratulations! Your manuscript is now with our production department. 

Kind regards, 

on behalf of

Dr. Christophe Hano 

Academic Editor

PLOS ONE